# Giant Triton Snail *Charonia tritonis* Macrophage-Expressed Gene 1 Protein Ct-Mpeg1: Molecular Identification, Expression Analysis, and Antimicrobial Activity

**DOI:** 10.3390/ijms232113415

**Published:** 2022-11-02

**Authors:** Wenguang Liu, Bing Liu, Gege Zhang, Gaoyou Yao, Yang Zhang, Xitong Cen, Maoxian He

**Affiliations:** 1CAS Key Laboratory of Tropical Marine Bio-Resources and Ecology, Guangdong Provincial Key Laboratory of Applied Marine Biology, South China Sea Institute of Oceanology, Chinese Academy of Sciences, Guangzhou 510301, China; 2Southern Marine Science and Engineering Guangdong Laboratory, Guangzhou 511458, China; 3University of Chinese Academy of Sciences, Beijing 100049, China

**Keywords:** *Charonia tritonis*, macrophage-expressed gene 1, antibacterial activity, expression pattern, *Vibrio alginolyticus*

## Abstract

Macrophage-expressed gene 1 proteins (Mpeg1/Perforin-2 (PRF2)) are a family of pore-forming proteins (PFPs) which can form pores and destroy the cell membrane of invading pathogens. However, little information is available regarding the function of Mpeg1 in the giant triton snail *Charonia tritonis*. In this study, a homolog of Mpeg1 (*Ct-Mpeg1*) was identified in *C. tritonis*. The predicted protein of *Ct-Mpeg1* contains several structural features known in Mpegs, including a membrane attack complex/perforin (MACPF) domain and single transmembrane region. The *Ct-Mpeg1* gene was constitutively expressed in almost all tissues examined except in the proboscis, with the highest expression level observed in the mantle. As a typical pore-forming protein, Ct-Mpeg1 has antibacterial activities against *Vibrio* (including *Vibrio alginolyticus* and *Vibrio parahaemolyticus*). In addition, rCt-Mpeg1 challenge to *V. alginolyticus* represses the expression of most outer membrane protein synthesis-related genes and genes involved in the TCA cycle pathway, which will lead to reduced outer membrane protein synthesis and less energy capacity. This is the first report to characterize the macrophage-expressed gene 1 protein in *C. tritonis*, and these results suggest that macrophage-expressed gene 1 protein Ct-Mpeg1 is an important immune molecule of *C. tritonis* that is involved in the bacterial infection resistance of *Vibrio*, and this study may provide crucial basic data for the understanding of the innate immunity system of *C. tritonis*.

## 1. Introduction

Macrophage-expressed gene 1 (Mpeg1/Perforin-2 (PRF2)) is an ancient metazoan protein belonging to the membrane attack complex/perforin (MACPF) branch of the MACPF/Cholesterol-Dependent Cytolysin (CDC) superfamily of pore-forming proteins (PFPs) [1]. Pore-forming proteins containing a MACPF domain are indispensable in innate immune defense because they form large transmembrane aqueous channels in target membranes of invading pathogens [2]. To date, three kinds of pore-forming molecules have been identified, members of the complement system (C6, C7, C8α, C8β, and C9), perforin 1, and perforin 2 [3,4,5]. Among them, perforin 2/Mpeg1, which also contains the MACPF domain [6], is a pore-forming, antibacterial protein with broad-spectrum activity [7]. When cells are infected by bacteria, perforin-2 vesicles translocate to and fuse with bacterium. Subsequently, perforin-2 polymerizes and forms large clusters of pores in the bacterial surface, making the bacteria susceptible to other bactericidal compounds such as lysozyme [7].

In contrast to vertebrates, possessing both the adaptive and innate immune systems which work together to detect and eliminate foreign organisms, invertebrates just possess innate defense systems against microbes and parasites [8]. Therefore, an effective protective mechanism is essential for the host′s survival to ward off detrimental foreign organisms. In terms of sponges, in *Suberites domuncula*, the oldest extant metazoan phylum, the homolog of Mpeg was identified as an immune effector used to eliminate Gram-negative and -positive bacteria, and its expression is largely upregulated after lipopolysaccharide (LPS) stimulation. In addition, the presence of the MACPF domain in combination with the increase in protein expression following exposure to bacterial PAMPs suggests that this ancient member of the perforin-2 family has potential antibacterial activity [9,10].

The upregulation of Mpeg1 expression following bacterial challenge is not exclusive to sponges. The increase in Mpeg1 caused by bacterial challenge has been demonstrated in several mollusks and echinoderms, such as abalone and oyster [11,12,13,14,15,16]. In addition, previous studies have shown that these external organs of marine invertebrates are primed to respond faster to foreign organisms than circulating hemocytes, possibly because they are frequently exposed to microbes in their environment, as evidenced in both abalone and oyster [13,14]. Furthermore, Mpeg1 was also proved to exhibit a dramatic growth suppression for Gram-positive as well as -negative pathogens, including Gram-negative bacteria *E. coli* and *V. alginolyticus* and Gram-positive bacteria *B. thuringiensis*, *S. aureus*, and *B. subtilis* [13]. Thus, Mpeg1 plays an important role in innate immunity in invertebrates.

The giant triton snail *C. tritonis* is an endangered gastropod species with ecological and economic importance [17]. Due to the filter feeding and sedentary habits of *C. tritonis*, it tends to be exposed to an aquatic environment that is rich in potential microbial pathogens. Similar to other invertebrates, *C. tritonis* have to solely rely on a strong innate immune defense system because they do not possess an adaptive immune system. In preliminary work, our team found the snails suffered from diseases and died when we conducted a breeding experiment on *C. tritonis*. Subsequently, several pathogenic bacteria, including *Vibrio* and *Staphylococcus,* were isolated and identified. Therefore, we speculated that the diseases of the snails caused by bacterial infection may possibly cause mass mortality in the future feeding process.

However, compared with the remarkable progress in vertebrates or other invertebrates, studies relating to *C. tritonis* Mpeg1 are scarce. We know little about the immune system of the giant triton snail, such as its immune molecules and immune mechanisms. Given the important roles that Mpeg1 plays in response to pathogen invasion, exploring novel family members and investigating the functional characteristics of *C. tritonis* are essential for understanding the immune mechanisms of Mpeg1 as well as exploring effective strategies and efficacious immunomodulatory methods to control bacterial diseases in this important species.

Here, we identified a homolog of Mpeg1 (Ct-Mpeg1) from the transcriptome of the giant triton snail (*Charonia tritonis*). To further explore their characterization and potential functions, we obtained one full-length open reading frame of the cDNA sequence of macrophage-expressed gene 1 *Ct-Mpeg1* and determined mRNA expression patterns of Ct-Mpeg1 in different tissues of *C. tritonis*. The recombinant Ct-Mpeg1 protein was expressed, and its antibacterial activity was characterized. In addition, the transcript abundances of the genes encoding key enzymes involved in the TCA cycles and the outer membrane protein of *V. alginolyticus* were quantified when challenged by rCt-Mpeg1. This is the first report to reveal the macrophage-expressed gene 1 protein in *C. tritonis*, which will provide crucial basic data for the understanding of innate immunity and the control of bacterial diseases in *C. tritonis*.

## 2. Results

### 2.1. Molecular Cloning and Bioinformatic Analysis of Ct-Mgep1 cDNA

Based on the gene annotation of the transcriptome of *C. tritonis* (NCBI sample ID: PRJNA695322) [17], we obtained the cDNA sequence of *Ct-Mgep1* (ID: CL636.Contig4_All), which was annotated as macrophage-expressed gene 1 protein. The cDNA sequence of the open reading frame of *Ct-Mgep1* was then determined by TA-cloning and deposited in Genbank under the accession number OP351737. As shown in Figure 1, the open reading frame (ORF) of *Ct-Mgep1* was 2355 bp (including stop codon) in size, and the theoretical molecular weight (MW) was calculated to be 85.75 KDa (Figure 1). 

A typical signal peptide of 28 a.a. residues was predicted by the signalP program in the N-terminus of Ct-Mgep1 (Figure 1). SMART program analysis revealed that the putative mature protein of *Ct-Mgep1* contained a membrane attack complex/perforin (MACPF) domain and single transmembrane region (Figure 2A). In addition, a 3-D model of the Ct-Mgep1 protein precursor was predicted using the SWISS-MODEL server (Figure 2B). Furthermore, a range of Mgep proteins from various species was collected to study the evolutionary relationships via phylogenetic analysis.

Blast analysis showed that the deduced amino acid sequences of Ct-Mgep1 share 67.86% and 56.46% identity, respectively, with the previously identified sequence from *Littorina littorea* and *Pomacea canaliculata*. Since Mgep1 is particularly well-defined in abalone (gastropod mollusks), it is worth mentioning that Ct-Mgep1 shares 51.53%, 51.25%, and 51.39% identity with three abalone species (*Haliotis madaka*, *Haliotis corrugata*, and *Haliotis discus discus*). Phylogenetic analysis was performed with several genes coding macrophage-expressed gene 1 proteins in different species. In this case, our newly identified Ct-Mgep1 has the shortest evolutionary distance from *Littorina littorea* (Figure 2C).

### 2.2. Expression Profiles of Ct-Mpeg1 mRNA in Different Tissues

The transcript expression of Ct-Mgep1 was detected in various tissues of *C. tritonis* via qPCR. As shown in Figure 3A, the expression of *Ct-Mgep1* could be ubiquitously detected in all tissues selected except in the proboscis, with the highest expression level in the mantle, followed by the tentacle, liver, and salivary glands. 

In addition, the transcriptome data before and after feeding showed that the highest expression level of *Ct-Mgep1* was seen in the liver among three tissues (digestive glands, liver, and salivary glands), followed by the digestive glands and salivary glands. However, the expression of *Ct-Mgep1* in the liver and digestive glands after feeding was at same level and also kept consistent with the transcript abundance before feeding. Meanwhile, the RPKM (Reads Per Kilobase per Million mapped reads) of *Ct-Mgep1* remained lowest in the salivary glands. It seemed that feeding had no effect on the expression of *Ct-Mgep1* in the digestive glands and salivary glands (Figure 3B). 

### 2.3. Prokaryotic Expression and Purification of rCt-Mpeg1

The ORF sequence of *Ct-Mgep1* was obtained from *C. tritonis* cDNA, and the sequence from +85 to +1086 bp (including the MACPF domain) was cloned into the pET-28b vector to produce rCt-Mgep1 in *E. coli* BL21 (DE3). rCt-Mgep1 was overexpressed after IPTG induction and subsequently purified by using Ni-NTA beads. SDS-PAGE analysis showed that the purified rCt-Mgep1 band was approximately 35 kDa (Figure 4A), which was in accordance with the Western blot band detected via the anti-6×His Tag antibody in Figure 4B.

### 2.4. The Antibacterial Activity of rCt-Mgep1 Protein

The inhibition effects of rCt-Mgep1 on the growth of *V. alginolyticus* (Figure 5A), *V. cholerae* (Figure 5B), *V. parahaemolyticus* (Figure 5C), *E. coli* (Figure 5D), and *S. aureus* (Figure 5E) were investigated. Obviously, *V. alginolyticus* exhibited a prolonged lag phase (>6 h) compared with the control group, implying that rCt-Mgep1 strongly inhibited the growth of *V. alginolyticus*. In addition, the growth of *V. parahaemolyticus* was weakly suppressed, as evidenced by a significant decreased maximum cell density. In contrast, the *S. aureus* strain had a reduced growth rate but unchanged maximum cell density. No significant difference in the growth curve of *E. coli* was exhibited, implying that rCt-Mgep1 showed no apparent inhibition effects to *E. coli*. Therefore, rCt-Mgep1 strongly suppressed the growth of the *V. alginolyticus* and *V. parahaemolyticus* bacteria. 

### 2.5. rCt-Mgep1 Has a Significant Impact on the Transcript Abundance of Genes Invloved in TCA Cycle and Outer Membrane Protein of V. alginolyticus

To better understand the mechanism responsible for the significant phenotypical changes observed in *V. alginolyticus*, the expression profiles of genes associated with the TCA cycle and outer membrane were selected (Table 1) and determined via qRT-PCR. The results are presented and discussed in categories based on their functions.

The expression profiles of *aceE*, *acnB*, *sucA*, *sucC* and *sdhA* involved in the TCA cycle were selected and measured (Figure 6). To some extent, all genes were downregulated, but at different times after rCt-Mgep1 addition. In detail, *aceE*, which encodes pyruvate dehydrogenase (acetyl-transferring), was downregulated at 10, 30, and 120 min (Figure 6A); *acnB*, which encodes bifunctional aconitate hydratase 2/2-methylisocitrate dehydratase, was downregulated at 60 and 120 min (Figure 6B); *sdhA*, which encodes succinate dehydrogenase catalytic subunit, was downregulated at 10, 30, 60, and 120 min (Figure 6E), while *sucA* and *sucC*, which encodes 2-oxoglutarate dehydrogenase, E1 component and succinyl-CoA synthetase subunit beta, respectively, showed merely decreased trends at 10 min after rCt-Mgep1 addition (Figure 6C,D). This suggests that rCt-Mgep1 may suppress the growth of *V. alginolyticus* by inhibiting its TCA cycle pathway at multiple points.

As is known, the membrane attack complex/perforin (MACPF) domains are adept in perforating and destroying the cytomembrane of invading pathogens [2]. To determine its effects on the outer membrane protein of *V. alginolyticus*, the expression profiles of *ompA*, *ompT*, *ompV*, *ompN1*, and *ompW*, which encode the outer membrane proteins, were measured. As shown in Figure 7, the expression of *ompV* was downregulated significantly at 10 and 30 min after rCt-Mgep1 addition; *ompF*, *ompT,* and *ompN1* were downregulated by 1.5~5-fold during protein challenge. This suggests that rCt-Mgep1 challenge to *V. alginolyticus* represses the expression of most outer membrane-protein-synthesis-related genes, leading reduced outer membrane protein synthesis.

## 3. Discussion

Although Mpeg1 has been widely studied in humans, only a few studies concerning mpeg1 have been carried out in mollusks and echinoderms, among which abalone and oyster have been most extensively explored [11,12,13,14,15,16]. However, little information is available to date regarding the functions of *C. tritonis* Mpeg1. In this study, we identified one homolog of Mpeg1 from an endangered species *C. tritonis* (Ct-Mpeg1) and determined the cDNA sequence of its ORF (open reading frame). The sequence contains a signal peptide, a conserved MACPF domain, a C-terminal transmembrane segment, and two low complexity regions. The MACPF domain is predicted to play a key role in defense against viral and bacterial pathogens [7,13,14,15], which suggests that *C. tritonis* Mpeg1 may also be critical for immune responses. 

To investigate the possible physiological functions of Mpeg1 in *C. tritonis*, we studied the mRNA levels in different tissues. Ct-Mpeg1 was expressed in almost all examined tissues except for the proboscis, and the highest expression was detected in the mantle and tentacle, which belong to external organs of *C. tritonis.* This implies that they respond effectively to foreign organisms because they are frequently exposed to microbes in their environment, which is consistent with the previous results in the Pacific oyster *C. gigas* and pink abalone *H. corrugate* [11,13]. In these two species, Mpeg1 has the highest expression in the gill, and this expression pattern contrasts intensively with that of human and mouse patterns or other vertebrate animals. Originally, the *Mepg1* gene was first characterized from a differential cDNA screen of human maturing macrophages. Additionally, initial studies demonstrated that it showed macrophage lineage-specific and mature stage-specific expression, but it was not expressed in other tissues or cell lines [18]. However, Podack et al. determined that *Mpeg1* is not exclusively localized to macrophages and can be identified in human and murine cell lines and primary cells following stimulation, which even can be induced to high expression levels following by IFN treatment or bacterial infection [6,19]. Such an expression pattern suggests that the *Mpeg1* gene is not only highly specialized and involved in the maturation or differentiation of mammalian macrophages, but also plays an important role in immune responses. It remains unknown whether Ct-Mpeg1 participates in the differentiation of hemocytes in *C. tritonis*. However, the constitutive expression of *Mpeg1* in invertebrates implies that it may have a more extensive role in innate immunity.

Given that *C. tritonis* is an endangered gastropod species and available samples are rare and quite precious, we did not perform a pathogen challenge assay to determine *Ct-Mpeg1* expression after bacterial tolerance. Nonetheless, we could speculate that the expression of Ct-Mpeg1 could be significantly upregulated after bacterial infection, as this is not the case in other species [6,19,20,21,22]. To further explore its potential antibacterial activity, recombinant Ct-Mpeg1 was constructed and purified. Our antibacterial activity assay has shown that rCt-Mpeg1 could strongly inhibit the growth of *V. alginolyticus* and *V. parahaemolyticus.* Thus, this evidence implies that Ct-Mpeg1 may have a crucial role in the elimination of invading pathogens in *C. tritonis*. 

*Vibrio* is widely distributed in the marine environment and is a common Gram-negative opportunistic pathogen present in a large variety of marine animals and human beings [23]. Here, we found Ct-Mpeg1 effectively suppresses the gene expression associated with the TCA cycle and outer membrane protein of *V. alginolyticus.* Carbon flux from the TCA cycle is crucial for the integration of carbohydrates with energy production and amino acid metabolism. So, it can be expected that the tolerance of rCt-Mpeg1 will lead to a deceleration of energy production, reducing the concentration of carbon intermediates that undergo amino acid biosynthesis de novo. Mpeg1 damages the cell walls of intracellular bacteria by insertion, polymerization, and pore formation of the perforin-like protein [24]. Outer membrane proteins of bacteria have also been described as the bacterial cytoskeleton, functioning to give bacteria their typical shape. The decreased expressions of outer membrane proteins suggest that rCt-Mpeg1 inserts into and perforates the cell membrane, and possibly also inhibits the expression of outer membrane proteins to promote immune responses.

In summary, one homolog *Mpeg1* molecule, named *Ct-Mpeg1*, was identified in *C. tritonis.* Except in proboscis, the *Ct-Mpeg1* gene was constitutively expressed in all tissues examined, with the highest expression level observed in the mantle. In addition, the recombinant Ct-Mpeg1 has been proven to have antibacterial activity against *V. alginolyticus* and *V. parahaemolyticus*. When challenged by rCt-Mpeg1, the expression of genes involved in the TCA cycle and outer membrane proteins were decreased significantly. To our knowledge, the present study represents the first discovery of the function of Mpeg1 in *C. tritonis*, thus contributing to a better understanding of the functional evolution of Mpeg1 in gastropods and providing basic data for the understanding of innate immunity and the control of bacterial diseases in *C. tritonis*.

## 4. Materials and Methods

### 4.1. Animals and Tissue Collection 

The adult *C. tritonis* were obtained from Nansha archipelagic waters of the South China Sea. Tissue samples (salivary glands, tentacle, proboscis, mantle, and liver) were separated and quickly frozen in liquid nitrogen for 24 h and then stored at 80 °C until total RNA extraction, as previously described by Zhang et al. [17].

### 4.2. Molecular Cloning and Bioinformatics Analysis of Ct-Mgep1

A full-length cDNA sequence of the *C. tritonis Ct-Mgep1* gene was found in a transcriptomic database constructed by our laboratory (GenBank ID: OP351737), and the open reading frame (ORF) sequence of *Ct-Mgep1* was obtained using the primers annotated *Ct-Mgep1*-F/R (Table 1). Extraction of total RNA and reverse transcription of first-strand cDNA were performed following the procedure described by Liu et al. [23]. The amino acid sequence alignment was performed with clustalx1.8, and the phylogenetic tree was built using the neighbor-joining method with 1000 bootstrap replicates using MEGA 6.0 [25] (accessed on 23 August 2022). The prediction of the structural domains of *Ct-Mgep1* was conducted with the SMART program [26] (accessed on 23 August 2022), and a three-dimensional (3-D) model was generated by using the SWISS-MODEL server [27,28] (accessed on 23 August 2022).

### 4.3. RNA Isolation and Quantitative Reverse Transcription PCR (qRT-PCR) Analysis

Multiple tissues, including salivary glands, tentacle, proboscis, mantle, and liver from adult *C. tritonis*, were examined. Total RNA was isolated using the TransZol Up Plus RNA Kit (TransGen Biotech, Beijing, China), and the Prime-Script RT Kit with gDNA Eraser (Takara Bio Inc., Kusatsu City, Japan) was used for reverse transcription. TB Green^®^ Premix Ex Taq™ II (Takara Bio Inc., Kusatsu City, Japan) was used for quantitative real-time PCR detection. The tissue expression pattern of Ct-PGRP-S1 was detected via quantitative real-time PCR (qPCR) using the gene-specific primers (Table 1), and 18s rDNA was used as an internal reference. 

For the total RNA isolation of *V. alginolyticus*, a strain was grown to 0.6 at 600 nm optical density, and 50 μg/mL recombinant Ct-Mgep1 protein was added to the culture with shaking. Cells were harvested immediately (t = 0) and at 10, 30, 60, and 120 min following the addition of recombinant Ct-Mgep1 protein and were put directly into liquid nitrogen. RNA was then purified from the samples as described above and used to generate cDNA. These genes, along with control 16S rRNA (Table 1), were detected via qRT-PCR. qRT-PCR analysis was carried out to verify gene expression as previously described [23]. Relative levels were calculated using the threshold cycle (ΔΔCT) method [29] and normalized to the wild-type *V. alginolyticus* value. Measurements were made in triplicate. Statistical significance was determined by the Student’s *t*-Test (ns *p* > 0.05, * *p* < 0.05, ** *p* < 0.01).

### 4.4. Construction of Prokaryotic Expression Vector

The construction of the prokaryotic expression vector was carried out as previously described [30]. Briefly, the cDNA segment (85–1086 bp) including the MACPF domain encoding a sequence of the *Ct-Mgep1* gene were amplified by PCR with the primer sets *Ct-Mgep1*-orf-F/R (Table 1). The plasmid pET28b was amplified with linearized primer pairs pET28b-F/R, and the fragments were then inserted into plasmid pET28b with a ClonExpress^®^ II One Step Cloning Kit (Vazyme Biotech Co., Ltd., Nanjing, China) to obtain a recombinant plasmid, which was transformed into *E. coli* BL21 (DE3)-competent cells. Subsequently, PCR and sequencing were used to check for the presence of the target genes with the primer pair pET28b-check-F/R.

### 4.5. Over-Expression and Purification of Recombinant Ct-Mgep1 Protein

The over-expression and purification of recombinant Ct-Mgep1 protein were carried out as previously described [30]. Bacterial Strain rCt-Mgep1-pET28b/BL21 (DE3) from a single colony was grown overnight in LB medium plus 50 μg of Kanamycin at 37 °C. A total of 500 μL of overnight *E. coli* BL21 (DE3) culture was diluted into 500 mL of fresh Luria broth (LB) with 50 mg/mL kanamycin (1:1000). The culture was incubated at 37 °C with shaking at 200 rpm until the cell density reached 0.6 at 600 nm optical density. The over-expression of recombinant proteins was induced by isopropyl-β-D-thiogalactoside (IPTG) at a 0.5 mM final concentration to 28 °C for 4 h. The induced cells were harvested and re-suspended in 10 mL of lysis buffer (50mM NaH_2_PO_4_, 300 mM NaCl, 5 mM imidazole, and 100 mM protease inhibitor PMSF). A pressure cell disruptor (Constant Systems, UK) was used for cell disruption, and cell debris was pelleted via centrifugation at 15,000× *g* for 30 min at 4 °C. The supernatant was incubated with 2 mL of nickel agarose beads for 3 h with rotation. The slurry was eluted by using a 50~500 imidazole gradient. The reaction mixture was then passed through a nickel agarose column, and purified protein was concentrated used Amicon Ultra centrifuge tube (Millipore, MA, USA) and stored in Tris buffer and stored in −80 °C. The purified rCt-Mgep1 protein was characterized using 12% sodium dodecyl sulfate–polyacrylamide gel electrophoresis (SDS-PAGE). Western blot was carried out according to the methods described by Liu et al. [30]. Briefly, the proteins were separated via SDS-PAGE and transferred to 0.2 μm polyvinylidene difluoride (PVDF) membranes (Millipore, MA, USA). The purified protein was detected using anti-6×His tag monoclonal antibody (Sangon Biotech, Shanghai, China), and SuperSignal™ West Pico PLUS (Thermo Fisher scientific, MA, USA) was used for visualization.

### 4.6. Antimicrobial Activity Assay

To investigate the antimicrobial activity of purified recombinant *Ct-Mgep1* protein, growth measurements in rich medium LB were carried out as previously described [23] with slight modification. In brief, the microbial strains used in the assays included Gram-negative bacteria *Escherichia coli, Vibrio alginolyticus*, and *Vibrio cholerae* and Gram-positive bacteria *Staphylococcus aureus*, grown overnight in LB medium at 37 °C. Each culture was brought to OD_600 nm_ = 1.0 using fresh LB and then diluted into fresh LB (1:300). The purified recombinant protein was diluted with sterile Tris (50 mM Tris-HCl, pH 8.0) to the final concentration of 50 μg/mL, and 50 μL of sterile Tris was added to 900 μL LB of *E. coli*, *V. alginolyticus*, *V cholerae, V. parahaemolyticus*, and *S. aureus* as the control. The bacterial growth kinetics of differenced bacteria were used to characterize the antibacterial activity of purified recombinant Ct-Mgep1 protein. Cultures (3 replicates in each case) were then incubated at 30 °C with continuous shaking at 200 rpm in 96-well plates. OD600nm was measured at regular time intervals using the Multiskan Ascent plate reader (Thermo Fisher Scientific, Waltham, MA, USA).

## Figures and Tables

**Figure 1 ijms-23-13415-f001:**
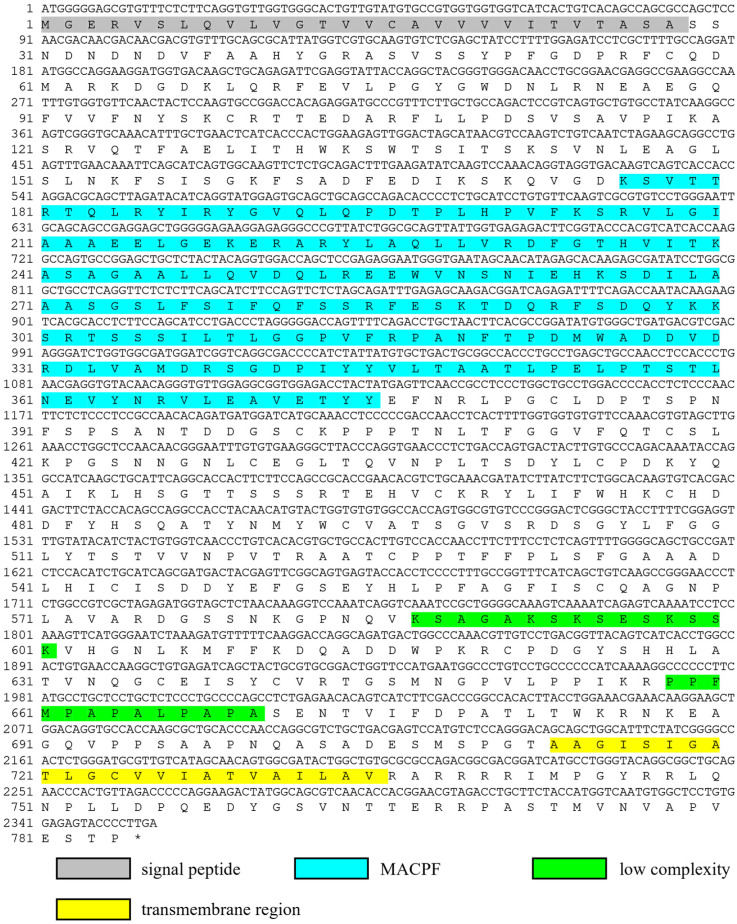
The full-length cDNA sequence of *Ct-Mgep1* and its deduced amino acid sequence. The signal peptide is shadowed; the MACPF domain is blue; the low complexity region is green; the transmembrane region is yellow.

**Figure 2 ijms-23-13415-f002:**
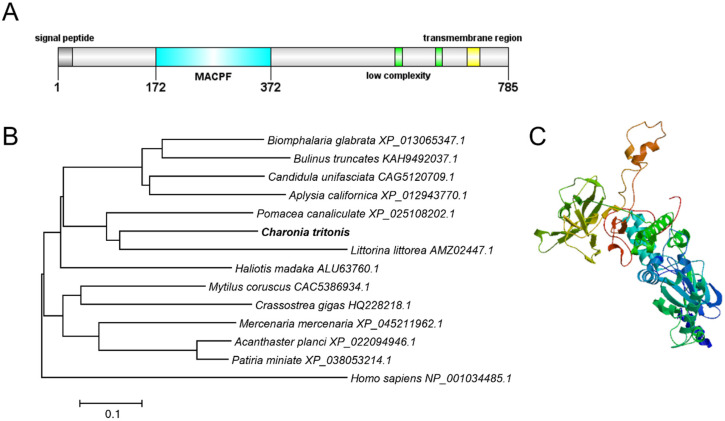
(**A**) Structural domain of Ct-Mpeg1 predicted by SMART program; (**B**) amino acid sequence alignment of Ct-Mpeg1 in multiple species. (**C**) Three-dimensional (3-D) protein model for Ct-Mpeg1 dimer by SWISS-MODEL server.

**Figure 3 ijms-23-13415-f003:**
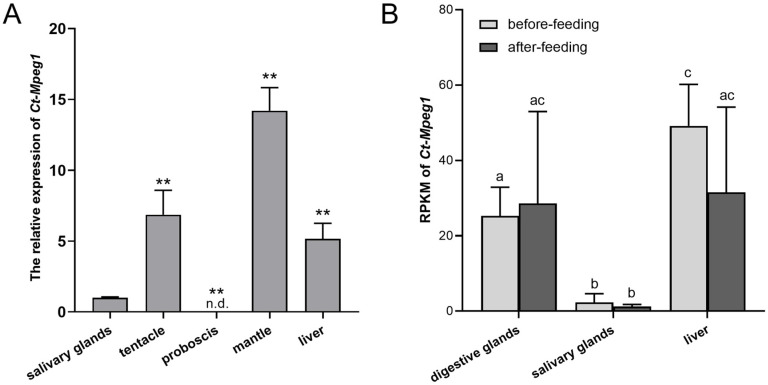
(**A**) Tissue distribution of *Ct-Mpeg1* mRNA. The selected tissues included the liver, proboscis, mantle, tentacle, and salivary glands. *Ct-Mpeg1* mRNA was highest in mantle, followed by tentacle, liver, salivary glands, and proboscis. Error bars indicate standard deviations (n ≥ 3). Significant differences between different groups are indicated with ** *p* < 0.01, n.d., not detected. (**B**) RPKM of *Ct-Mpeg1* mRNA in three tissues (digestive glands, liver, and salivary glands) before and after feeding. After feeding, levels remained same in liver, digestive, and salivary glands. Means not sharing the same superscript are significantly different in each tissue sample. Error bars indicate standard deviations (n = 3).

**Figure 4 ijms-23-13415-f004:**
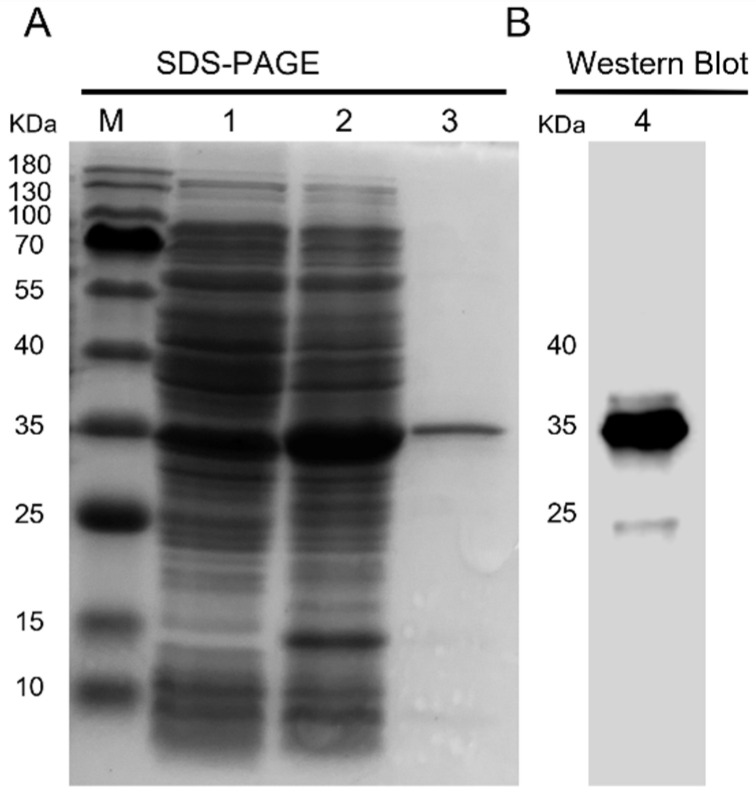
Prokaryotic expression and Western blot analysis of r Ct-Mgep1. (**A**) The SDS-PAGE analysis of prokaryotic-expressed rCt-Mgep1. Lane M: protein marker; 1 and 2: protein from BL21 (DE3) transformed with pET-28b-rCt-Mgep1 plasmid before and after 0.5 mM IPTG induction; 3: purified rCt-Mgep1. (**B**) Western blot analysis of rCt-Mgep1 with anti-6×His tag monoclonal antibody.

**Figure 5 ijms-23-13415-f005:**
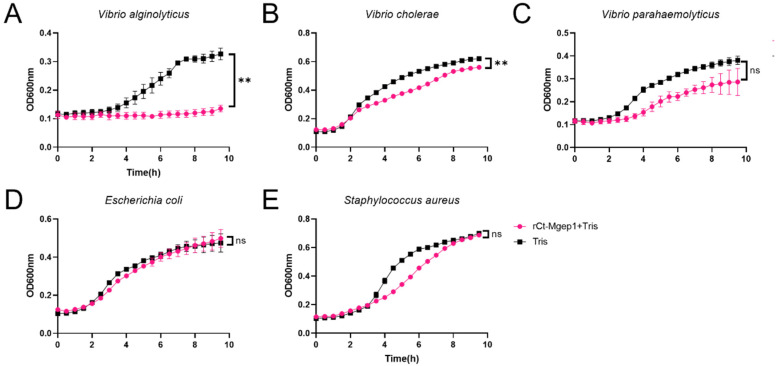
Growth suppressive test of rCt-Mgep1 against bacteria. *V. alginolyticus* (**A**), *V. cholerae* (**B**) *V. parahaemolyticus* (**C**), *E. coli* (**D**), and *S. aureus* (**E**) *were mixed with* rCt-Mgep1 (final concentration 50 μg/mL) and the OD_600nm_ was recorded per half an hour, lasting for 10 h. For growth curves, three biological replicates are shown as points with their average values connected by lines. Significant differences in maximum cell density between different groups are indicated with ** *p* < 0.01, ns, not significant. Error bars indicate the standard error of the mean (SEM).

**Figure 6 ijms-23-13415-f006:**
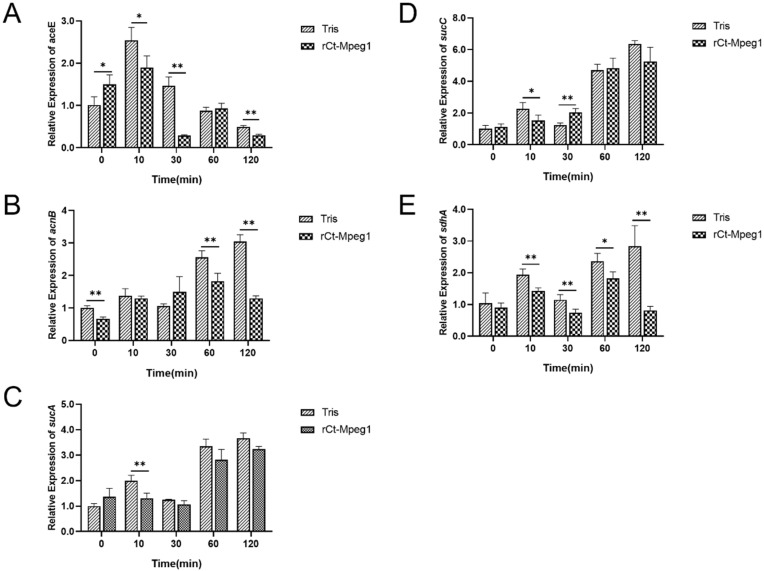
rCt-Mgep1 affects the transcript abundance of genes involved in TCA cycle of *V. alginolyticus*. The relative expression of *aceE* (**A**), *acnB* (**B**), *sucA* (**C**), *sucC* (**D**), *sdhA* (**E**). The relative expression level of a set of genes was obtained for each strain from the integration of three biological replicates (see “Section 4” for details). Data are presented as the mean ± SD; * *p* < 0.05, ** *p* < 0.01.

**Figure 7 ijms-23-13415-f007:**
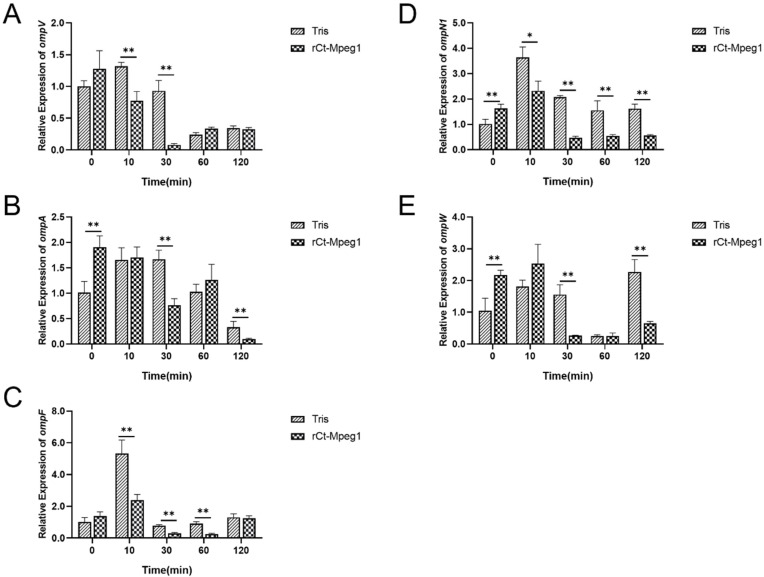
rCt-Mgep1 affects the transcript abundance of genes involved in outer membrane proteins of *V. alginolyticus*. The relative expressions of *ompV* (**A**), *ompV* (**B**), *ompF* (**C**), *ompN1* (**D**), and *ompW* (**E**) were obtained for each strain from the integration of three biological replicates (see Section 4 for details). Data are presented as the mean ± SD; * *p* < 0.05, ** *p* < 0.01.

**Table 1 ijms-23-13415-t001:** Nucleotide sequences of primers used in this study.

Primers	Sequence (5′-3′)
For ORF cloning	
*Ct-Mgep1*-F	GAGAAAGTCGCACGAAAGTTG
*Ct-Mgep1*-R	GGATCGTCAATCATAACAATTC
For qPCR	
Q*Ct-Mgep1*-F	CTCCGCCAACACAGATGATG
Q*Ct-Mgep1*-R	TGGAAGAAGTGGTGCCTGAA
Q18S-F	ATGGTCAGAACTACGACGGTAT
Q18S-R	GTATTGCGGTGTTAGAGGTGAA
Q-*aceE*-F	TTTCCAGTCTTCTGCTGCGT
Q-*aceE*-R	CATCAGTTTAGGGTGCGGGT
Q-*acnB*-F	TTCAACGCGTAACTTCCCGA
Q-*acnB*-R	CGTAGCGTCGATTTGCTTCG
Q-*sucC*-F	AGCATCCTACACCACGCAAA
Q-*sucC*-R	AGTCTACAGAGTGCAACGCC
Q-*sucA*-F	ACCAAACTGACGCTAACGGT
Q-*sucA*-R	GGCGTTTCTTCCGCAACTTT
Q-*sdhA*-F	ATGCGTTGGGAAAACAGCAC
Q-*sdhA*-R	AGACTTGTCCGCTAGATGCG
Q-*ompA*-F	ACGTTGGTGGTAAGATGGGT
Q-*ompA*-R	TGGCCACCGAAAGAAGTTTG
Q-*ompT*-F	CAAGCGAGCGTACAGATGAC
Q-*ompT*-R	ATCGTTGTTCCAAGCGTGAC
Q-*ompV*-F	TGGAACGTAGAAGCTGGTGT
Q-*ompV*-R	TTTTCGCTGTATCACCGTCG
Q-*ompN1*-F	CCTTGCTGCGGTTTATGGTT
Q-*ompN1*-R	GTGCCATTTCTTCACGAGCA
Q-*ompW*-F	TAGCAGCAACACCATTCAGC
Q-*ompW*-R	TGAGACCTGCACCAACGTAT
Q16S-F	CTGGAACTGAGACACGGTCC
Q16S-R	CTCGCACCCTCCGTATTACC
For Ct-Mgep1 recombinant protein construction	
pET28b-F	CACCACCACCACCACCAC
pET28b-R	GGTATATCTCCTTCTTAAAGTTAAACAAAATTATTTC
*Ct-Mgep1*-orf_F	ctttaagaaggagatatacATGAGCTCCAACGACAACG
*Ct-Mgep1*-orf_F	cagtggtggtggtggtggtgCTCGTTCAGGGTGGAGGTTG
pET28b-check-F	AAGTGGCGAGCCCGATCTTC
pET28b-check-R	CTAGGGCGCTGGCAAGTGTA

## Data Availability

The mRNA sequence of Ct-Mpeg1 and the transcriptome of the giant triton snail (*C. tritonis*) were deposited in GenBank. The accession number for Ct-Mpeg1 is GenBank ID: OP351737; the sample ID transcriptome is PRJNA695322. Other data presented in this study are available in the article.

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
