# Peer review of "Giant Triton Snail Charonia tritonis Macrophage-Expressed Gene 1 Protein Ct-Mpeg1: Molecular Identification, Expression Analysis, and Antimicrobial Activity"

_ijms, 2022, doi:10.3390/ijms232113415_

Round 1

Reviewer 1 Report

Review for: “Giant triton snail Charonia tritonis Macrophage-expressed gene 1 protein Ct-Mpeg1: molecular identification, expression analysis, and antimicrobial activity”

In the present study, the authors identified and characterized a homolog of Mpeg1 in C tritonis, and analyzed its expression in different C. tritonis tissues and investigated in vitro its antimicrobial activity against Vibrio and Staphylococcus aureus. While the authors argue that the information gathered in this manuscript is crucial to the understanding of innate immunity and control of bacterial diseases in C. tritonis, the lack of data, especially concerning C tritonis, does not help reach such conclusions. Moreover, an additional purification step must be done prior to any further conclusion

Major observations:

  1. The presented data is not sufficient to make any conclusions regarding the innate immune system and bacterial diseases in C. tritonis. Other than data that suggest that Ct-Mpeg1 is expressed in several tissues, there are no pathogens challenge assay or any other studies (i.e loss or gain or function, competition, etc) that suggest it plays an important role in the innate immune system of C. tritonis. While the authors point out that samples are rare and precious and that is the reason why such studies were not performed it is an overstatement to say that it is crucial for the immune system
  2. Lack of important information concerning the qPCR data and analysis. The authors wrote “18s rDNA was used as an internal reference.”, what does this mean? Typically gene expression analysis by qPCR is followed by a ΔΔCt analysis and authors should indicate specifically whether they performed any gene expression analysis and what the analysis was. Secondly, I would like the authors to indicate what was the expression in each gene compared to the housekeeping because otherwise, it is impossible to understand whether the gene is very or lowly expressed in the tissue.
  3. In Figure 2, the authors indicate that Ct-Mgep1 share 67.86% and 56.46% with Littorina littorea and Pomacea canaliculata. I would have expected a higher homology to both species (~90%). Why is homology so low? Also, it would be interesting to see in Figure 2 the percentage for all specie.
  4. In Figure 3, what is the difference between the data from panel A and the before feeding in panel B? Shouldn’t it be the same? Also, why are the relative expression values so different between A and B
  5. In Figure 4, the western blot shows a clear band around the 35 KDa but previously the authors stated that the protein was predicted to be approximately 75KDa. How do they explain it?
  6. In Figure 5, no statistical analysis was performed. This is important because otherwise, the authors cannot make any conclusions.
  7. Purification of recombinant Ct-Mgep1 protein. My major issue with this study was the usage of the recombinant protein prior to any additional chromatography. Ultimately, the recombinant protein is full of contaminants whose effect is currently not clear. For instance, it is not clear that this effect is not due to diluted antibiotics. While I strongly advise an additional purification step the authors should have used a negative control in which they should have used extracts from bacteria using a control plasmid.

Author Response

Dear reviewers:

Re: Manuscript ID: ijms-1922209 and Title: Giant triton snail Charonia tritonis Macrophage-expressed gene 1 protein Ct-Mpeg1: molecular identification, expression analysis, and antimicrobial activity.

Thank you for your letter and the reviewers’ comments concerning our manuscript entitled “Giant triton snail Charonia tritonis Macrophage-expressed gene 1 protein Ct-Mpeg1: molecular identification, expression analysis, and antimicrobial activity” (ijms-1922209). Those comments are valuable and very helpful. We have read through comments carefully and have made corrections. Based on the instructions provided in your letter, we uploaded the file of the revised manuscript. Revisions in the text are shown using Yellow highlight. The responses to the reviewer's comments are marked in red and presented following.

我们非常感谢您允许我们重新提交手稿的修订副本,我们非常感谢您的时间和考虑。如果您有任何疑问,请毫不犹豫地与我们联系。

真诚地。

刘兵

Reviewer 2 Report

Dear authors, 

Congratulations on the current research project. The manuscript reads well and I have a few suggestions which are enumerated as follows

1. Line 135: change to "liver, and salivary glands"

2.  Figure 3 Legend: i. Highlight the title of the legend

ii. Please explain the figures well in legends. For example, Ct-Mpeg1 mRNA was highest in mantle, followed by tentacle, liver, and salivary glands. Also add that after feeding, levels were reduced in liver while remaining same in digestives and salivary glands.

iii. Replace "The data are presented as the means  SD (n ≥ 3)" with tentacles, and salivary glands; n>3 (mean+S.D).

iv. Replace "Significant differences between different groups are indicated with an asterisk at p < 0.05, and with two asterisks at p < 0.01." with the following "*p<0.05; **p<0.01."

3. Line 148: Did the authors compare the data statistically, before and after feeding? If yes, please mention it. If not, please check the results and mention them appropriately.

4. Figure 4B: Please provide size marker for western blot image.

5. Highlight/ Bold the heading of the figure legend. The first word of a sentence should be capital "Prokaryotic expression and western blot analysis of r Ct-157 Mgep1"

6. Highlight/ bold the headings of all figure legends.

Thanks!

Author Response

(The authors gave the same response as above.)

Round 2

Reviewer 1 Report

Most of the questions and issues were adressed

Reviewer 2 Report

Dear Authors,

Thank you so much for providing the revision. Following are my suggestions, please consider them for further revision.

1.       Please provide a full gel picture of the western blot for figure 4 with a size marker. Without an appropriate size marker in the same gel picture, the results are not acceptable.

2.       When mentioned to highlight the title of the figure legend, it should done as follows:

‘Figure 5. Growth suppressive test of rCt-Mgep1 against bacteria. V. alginolyticus (A), V. cholerae (B) 176 V. parahaemolyticus (C), E. coli (D), and S. aureus (E) were mixed with rCt-Mgep1 (Final concentration 177 50 μg/mL) and the OD600nm was recording per half an hour lasting for 10 h. For growth curves, three 178 biological replicates are shown as points with their average values connected by lines. Significant 179 differences of maximum cell density between different groups are indicated with **p<0.01, ns., not 180 significant. Error bars indicate the standard error of the mean (SEM). >’

Author Response

Dear reviewer:

Re: Manuscript ID: ijms-1922209 and Title: Giant triton snail Charonia tritonis Macrophage-expressed gene 1 protein Ct-Mpeg1: molecular identification, expression analysis, and antimicrobial activity.

Thank you for your letter again and the reviewer’s comments concerning our manuscript entitled “Giant triton snail Charonia tritonis Macrophage-expressed gene 1 protein Ct-Mpeg1: molecular identification, expression analysis, and antimicrobial activity” (ijms-1922209). Those comments are valuable and very helpful. We have read through comments carefully and have made corrections. Based on the instructions provided in your letter, we uploaded the file of the revised manuscript. Revisions in the text are shown using Red highlight. The responses to the reviewer's comments are marked in red and presented following.

We would love to thank you for allowing us to resubmit a revised copy of the manuscript and we highly appreciate your time and consideration. Should you have any questions, please contact us without hesitation.

Sincerely.

Bing LIU

Comments and Suggestions for Authors

Dear Authors,

Thank you so much for providing the revision. Following are my suggestions, please consider them for further revision.

  1. Please provide a full gel picture of the western blot for figure 4with a size marker. Without an appropriate size marker in the same gel picture, the results are not acceptable.

Response 1: Thank you for your helpful advice. But we feel sorry about that. When we did western blotting, the size marker doesn't show up on the figure during development because it does not have horseradish peroxidase or biotin (coupled to streptavidin).

  1. When mentioned to highlight the title of the figure legend, it should done as follows:

“图 5。rCt-镁十一酮对细菌的生长抑制试验。将溶藻弧菌(A)、霍乱弧菌(B)176副溶血性弧菌(C)、大肠杆菌(D)和金黄色葡萄球菌(E)与rCt-Mgep1(终浓度177 50μg/mL)混合,每半小时记录OD600nm持续10小时。对于生长曲线,三个178个生物重复显示为点,其平均值由线连接。不同组之间最大细胞密度的显著差异为179,用**p<0.01,ns.表示,而不是180显著。误差线指示均值 (SEM) 的标准误差。>'

回应2:感谢您的建议。但是,不应根据 IJMS 作者说明突出显示图形图例的标题。我们对此感到抱歉。

Round 3

Reviewer 2 Report

Dear authors, 

Thank you for your honest response. While the authors were granted major revision which includes extended time, they have been unable to provide me with the original image. I have to, unfortunately, reject this manuscript.